# Prevalence of Peripheral Arterial Disease and Associated Vascular Risk Factors in 65-Years-Old People of Northern Barcelona

**DOI:** 10.3390/jcm10194467

**Published:** 2021-09-28

**Authors:** Gabriela Gonçalves-Martins, Daniel Gil-Sala, Cristina Tello-Díaz, Xavier Tenezaca-Sari, Carlos Marrero, Teresa Puig, Raquel Gayarre, Joan Fité, Sergi Bellmunt-Montoya

**Affiliations:** 1Angiology, Vascular and Endovascular Surgery Department, Hospital Vall d’Hebron, 08035 Barcelona, Spain; ggoncalves@vhebron.net (G.G.-M.); cristina.tello11@gmail.com (C.T.-D.); atenezaca@vhebron.net (X.T.-S.); cmarrero@vhebron.net (C.M.); sbellmunt@vhebron.net (S.B.-M.); 2Departament de Cirurgia, Universitat Autònoma de Barcelona (UAB), 08035 Barcelona, Spain; 3Department of Clinical Epidemiology and Public Health, Hospital de la Santa Creu i Sant Pau. IIB Sant Pau, 08041 Barcelona, Spain; tpuig@santpau.cat; 4Deparment of Pediatrics, Obstetrics and Gynecology and Preventive Medicine, Universitat Autònoma de Barcelona (UAB), CIBERCV. 08035, Barcelona, Spain; 5Primary Care Department, Institut Català de la Salut, 08041 Barcelona, Spain; rgayarre@gencat.cat; 6Departament de Medicina, Unitat Docent Sant Pau, Universitat Autònoma de Barcelona (UAB), 08025 Barcelona, Spain; 7Angiology, Vascular and Endovascular Surgery Department, Hospital de la Santa Creu i Sant Pau, Institut de Recerca Biomèdica de Sant Pau (IIB-Sant Pau), CIBERCV, 08041 Barcelona, Spain; jfite@santpau.cat

**Keywords:** peripheral arterial disease, cardiovascular risk factors, asymptomatic, ankle–brachial index, prevalence, screening

## Abstract

Objective: To determine the prevalence and risk factors associated with peripheral arterial disease (PAD) in Northern Barcelona at 65 years of age. Methods: A single-center, cross-sectional study, including males and females 65 years of age, health care cardholders of Barcelona Nord. PAD was defined as an ankle–brachial index (ABI) < 0.9. Attending subjects were evaluated for a history of common cardiovascular risk factors. A REGICOR score was obtained, as well as a physical examination and anthropometric measurements. Results: From November 2017 to December 2018, 1174 subjects were included: 479 (40.8%) female and 695 (59.2%) male. Overall prevalence of PAD was 6.2% (95% CI: 4.8–7.6%), being 7.9% (95% CI: 5.9–9.9%) in males and 3.8% (95% CI: 2.1–5.5%) in females. An independent strong association was seen in male smokers and diabetes, with ORs pf 7.2 (95% CI: 2.8–18.6) and 1.8 (95% CI: 1.0–3.3), respectively, and in female smokers and hypertension, with ORs of 5.2 (95% CI: 1.6–17.3) and 3.3 (95% CI: 1.2–9.0). Male subjects presented with higher REGICOR scores (*p* < 0.001). Conclusion: Higher-risk groups are seen in male subjects with a history of smoking and diabetes and female smokers and arterial hypertension, becoming important subgroups for our primary healthcare centers and should be considered for ABI screening programs.

## 1. Introduction

Peripheral arterial disease (PAD) is the manifestation of atherosclerotic disease in the lower extremities, resulting in narrowing of the blood vessels and diminishing blood flow to the limbs. It is the third leading cause of cardiovascular pathology after coronary and cerebrovascular disease and an important indicator of cardiovascular risk [1]. The mere presence of PAD can indicate a 6.6-fold increased risk of cardiovascular disease association [2], becoming, therefore, an important indicator of the coexistence of other cardiovascular pathologies.

Associated risk factors are common for cardiovascular diseases, such as smoking, diabetes, dyslipidemia and hypertension, among others. Active smoking has the strongest association, doubling the risk of PAD in these patients [3]. Therefore, early diagnosis and treatment of modifiable risk factors is an important tool in preventing and treating this disease, consequently lowering associated cardiovascular disease morbidity and mortality.

The ankle–brachial index (ABI) is an effective, easy and first-line noninvasive method in screening PAD [4,5,6], with an overall 69–79% sensitivity and 83–99% specificity for detecting >50% arterial stenosis [7]. Despite the feasibility of this technique, this pathology is well known to be underdiagnosed and undertreated [4,5].

In the United States, 8 to 12 million Americans suffer from PAD, with an overall prevalence of 3–10%, which increases to almost 50% in those greater than 85 years old [8,9]. As for Europe, prevalence as high as 17.8% (99% CI: 16.84–18.83%) was observed in some studies, especially in the northern region [10].

PAD is also influenced by gender and age, predominantly being associated with the male gender. Age is also an important factor, with a 20% increase observed in those older than 75 [11,12]. In a global aging society, this can present an important social and economic burden.

To our knowledge, most studies focused on the prevalence of PAD performed in our region were inclusive of younger age groups, some including ages as young as 35 years old [13,14]. This perhaps explains the low prevalence results achieved in these studies. Due to the important variety of factors that influence the prevalence of this pathology, age being one of them, we decided to perform a study including individuals of 65 years old in order to analyze if an increase of prevalence was seen in these groups and, as such, evaluate if there were differences in our gender groups with this increase in age, as well as associated risk factors. This way perhaps allows pinpointing a target group that requires a more detailed initial evaluation and opportunistic screening (with ABI) in our primary care centers in order to prevent and/or diagnose PAD.

Thus, the aim of our study was to determine the prevalence of peripheral arterial disease through ABI in 65-year-old men and women in Northern Barcelona (Spain), as well as its associated risk factors.

## 2. Materials and Methods

This is a single-center, populational-based, cross-sectional study, part of a larger pilot screening program evaluating abdominal aorta aneurysm [15]. The protocol was approved by the ethics committee of Hospital Vall d’Hebron with code PR(AG)221/2017.

### 2.1. Patient Selection

The study was carried out in a health-integrated area AIS-Barcelona Nord, with a population base of 400,000 inhabitants, which corresponds to the area of treatment of our center. Inclusion criteria were all noninstitutionalized males and females of 65 years of age during the time of recruitment who were health care cardholders and resided for at least 6 months in the referred area.

Contact information of the population was authorized and obtained through our public health census registry, and all eligible subjects received invitation letters informing them of the present study and tests to be performed. Those with incorrect postal data, terminal disease and deceased subjects were considered ineligible (Figure 1).

Nonresponders to the first letter were invited again within 1 month. Those who agreed to participate were scheduled an appointment at our Vascular Lab Center with one of five trained vascular surgeon residents. Participants were contacted 48 h before the appointment to confirm attendance. Those who did not show up for the scheduled appointment after 3 occasions, those who did not wish to participate and subjects who did not respond to both invitation letters were excluded.

### 2.2. Variables Evaluated

After arrival, prior to questioning, a signed consent form was required from each participant. All subjects were questioned for a history of risk factors, such as: cigarette smoking: active, former (more than a year without smoking) or nonsmoker; hypertension, defined as systolic blood pressure ≥140/90 mmHg or subjects under antihypertensive treatment; dyslipidemia, defined as one or all of total blood cholesterol >200 mg/dL, low-density lipoprotein (LDL) cholesterol >100 mg/dL or triglycerides >150 mg/dL or subjects under treatment for dyslipidemia; diabetes, defined as those diagnosed and/or under antidiabetic treatment; history of cerebrovascular disease (previous stroke or transient ischemic cerebral accident); history of cardiac ischemia (diagnosed coronary artery disease or previous myocardial infarction); chronic renal disease (glomerular filtration <60 mL/min) and history of any aortic aneurismatic disease. Anthropometric measurements were also recorded: height (m), weight (kg), waist circumference (cm) and body mass index (BMI). A REGICOR score [16] was determined for each individual to evaluate primary prevention, discarding those subjects who had already presented a previous cardiovascular event. Cholesterol levels and blood pressure required for scoring were obtained through medical records. The REGICOR (Girona Heart Registry) research group focuses on ischemic heart disease and associated risk factors in order to improve preventive strategies. This score assesses the risk of presenting a cardiovascular event in the following 10 years through a computer-generated calculation that evaluated age, sex, history of smoking, presence of diabetes and blood pressure, total cholesterol and high-density lipoprotein (HDL) levels. Low risks are considered to be scores <5%, moderate risk between 5 and 9% and high-risk subjects >10%.

### 2.3. Physical Assessment

To complete the assessment, a physical exam was performed, evaluating distal pulses through palpation. The presence of a pulse in one or both tibial arteries was considered adequate. Afterward, an ABI was performed in a standard manner [7]: after a 5 min rest, with the subjects in a supine position, systolic blood pressure was measured at the level of the posterior and anterior tibial arteries of both lower extremities and at the level of the brachial artery of both upper extremities, with a continuous wave Doppler probe (Flowsoft 7 Angiolab1 Spead Doppler Systeme, Kehl, Germany) and an arterial pressure cuff that was placed just above the malleoli. ABI was calculated through the ratio of systolic pressure of the tibial arteries to the highest brachial systolic pressure. ABI of both extremities was recorded. Those with an ABI < 0.9 in one or both lower extremities were considered pathological with an indication of PAD. All ABI > 1.4 were registered. In the case that a patient presented with critical limb ischemia, considered as rest pain or presence of ischemic lesions, they were sent to the vascular emergency room for evaluation. At the end of each assessment, the patient was given a signed report indicating the result of their ABI exam, and if the exam was pathological, recommendations of the modifiable risk factors were given in order to prevent the progression of PAD. 

For better analysis and interpretation of our results, we reviewed common risk factors among our study participants and those obtained from the official health department database, 2018 Catalan Health Survey “Enquesta Catalunya de Salut 2018” (ESCA) (ages 65–74 years old), and our primary care clinics (65 years old), with previous authorization from our health department.

### 2.4. Statistical Analysis

Descriptive analysis was performed, and prevalence of PAD was obtained with the 95% confidence interval. Dichotomous variables were analyzed through Pearson’s chi-square test or Fisher’s exact test. Logistic regression models were used to obtain independent associated risk factors for PAD that were previously significant in the bivariate analysis. Interaction effect was also evaluated among these risk factors. Statistical significance results were considered for a value of *p* < 0.05. Statistical analysis was performed through IBM SPSS Statistics for Windows, version 22.0 (IBM Corp., Armonk, NY, USA).

## 3. Results

From November 2017 to December 2018, a total of 2808 subjects were 65 years old during the time of recruitment. A total of 1174 subjects were finally included, 695 (59.2%) male and 479 (40.8%) female. Total participation rate was 43.1%, with higher participation from females (46.7%) than males (40.9%). A flow chart of initial screening with main results separated by sex is shown in Figure 2.

The overall prevalence of PAD was 6.2% (95% CI: 4.8–7.6%), male 7.9% (95% CI: 5.9–9.9%) and female 3.8% (95% CI: 2.1–5.5%), most of them corresponding to the Fontaine I classification (*n* = 53, 72.6%), 20.5% IIA (*n* = 15), 5.5% IIB (*n* = 4) and 1.4% (*n* = 1). The difference in prevalence between male and female was statistically significant (*p* = 0.006). Male subjects presented at higher risk for PAD, with an OR of 2.2 (CI: 1.3–3.8).

There was a total of 18 ABI > 1.4, of which two of these had contralateral ABIs < 0.9 considered PAD. Of the other 16 ABIs > 1.4 (6 were diabetic subjects, and none had chronic renal disease), all subjects except for one had palpable distal pulses and none referred to intermittent claudication.

Diabetes male subjects were more likely to have PAD (*p* = 0.017) and female subjects hypertension (*p* = 0.026) and waist circumference (*p* = 0.018). No association was seen with a family history of abdominal aortic aneurysm, probably due to the small number of cases detected. Table 1 compares risk factors presented in PAD and non-PAD divided by gender.

A multilogistic regression model was performed from the significant bivariate analysis variables. In our male model, an independent association between PAD and smoking was shown. PAD in male subjects was higher in ex-smokers, with an OR of 3.5 (95% CI: 1.4–8-7) and in active smokers 7.2 (95% CI: 2.8–18.6) in comparison with nonsmokers. An association with diabetes and PAD was also seen in our male participants, with an OR of 1.8 (95% CI: 1.0–3.3)

As for our female model, an association was also seen in relation to PAD and smoking, with an OR of 3.1 (95% CI: 1.0–9.6) in ex-smokers and 5.2 (95% CI: 1.6–17.3) in active smokers compared with nonsmokers. PAD was also higher in those female subjects with hypertension, with an OR of 3.3 (95% CI: 1.2–9.0) (Table 2). No significant association was seen between hypertension and PAD in male subjects nor diabetes and PAD in our female subjects. Therefore, no interaction effect was observed among these variables, and no mediation effect was observed with gender.

Discarding 157 subjects who had already presented a previous cardiovascular event, 1013 subjects were classified according to their cardiovascular risk at 10 years with the REGICOR scale. In this regard, 51.4% (*n* = 500) of our population were at low risk of developing a cardiovascular event in the following 10 years, 40.4% (*n* = 393) were at moderate risk and 8.1% (*n* = 79) were at a high-risk score. Four subjects were excluded from REGICOR analysis due to missing data. Male subjects presented with higher REGICOR scores (*p* < 0.001), putting them at a higher risk for developing a cardiovascular event in the following 10 years compared to our female subjects. Table 3 illustrates the REGICOR score in our general population divided by sex and PAD.

Characteristics of our 1174 studied men and women are best shown in Table 4. Overall, fewer women were smokers (*p* < 0.001), and our male subjects had higher percentages of diabetes (*p* < 0.001), high blood pressure (*p* < 0.001), chronic renal insufficiency (*p* = 0.001) and cardiac ischemia (*p* < 0.001), presenting with more risk factors than our female participants. Regarding the history of aneurysmal disease, 3% (*n* = 35) had a family history of abdominal aneurismatic disease, with 26 of them being a first-degree family member.

Characteristics of our participants with those obtained from the 2018 Catalan Health Survey “Enquesta Catalunya de Salut 2018” (ESCA) and primary care centers located in our area of treatment of Barcelona Nord were reviewed in order to better assess the representation of our sample participants with our overall area of treatment population, as shown in Table 5.

## 4. Discussion

In the present study, the prevalence of PAD in asymptomatic patients was 6.2% at age 65, which is in accordance with previous studies carried out in the Mediterranean area [13,14,17]. Risk factors are commonly known to be associated with other atherosclerotic diseases, with predominant factors being those of active smoking and diabetes [14]. This was shown to be similar in our study, with associated risk factors being smoking, history of diabetes, cardiac ischemia and a large waist circumference, smoking being the strongest association. Independent risk factors were also seen in logistic regression analysis in male smokers who had a history of diabetes and in female smokers who had hypertension. Knowing the important association of PAD with high cardiovascular risk groups, nonscreening in these subjects could lead to an underdiagnosis or even underestimate the true cardiovascular risk of this disease, outlining the importance of screening in these subgroups.

### 4.1. Differences According to Geography and Age

The worldwide prevalence of PAD is estimated to be 3–12% [18], affecting 27 million people in America and Europe [19]. In Europe, the prevalence of PAD is estimated at around 17.8% (99% CI: 16.8–18.8%) in the ages of 45 and 55, seen in the PANDORA study [10].

In Spain, however, the prevalence of PAD is considerably lower. The ESTIME study [20] estimated an 8.03% prevalence in similar age groups as the PANDORA study [10] in Europe, and studies published in the Catalonia Region of Spain concur with these low results. In this sense, Velescu et al. [14] reported a prevalence as low as 4.5% in Girona, and the Perart/Artper group of Barcelona reported a 7.6% prevalence [17]. Although both results are relatively much lower than estimated in Europe and the USA, the observed difference amongst the two might be due in part to the difference in the studied age group, with the Velescu group including subjects as young as 35 years old. This demonstrates the increase in prevalence with age that has already been described previously and the variability of prevalence with geographic regions due to ethnicity and dietary lifestyle, which tend to be unique to each region of study [21,22].

### 4.2. Sex Differences

Traditionally PAD is thought to be a male-predominant disease [23], but in recent studies, this has been proven otherwise, with an increase of prevalence in elderly women. About 20–30% of women over 70 years old are affected by PAD [24]. Peripheral arterial disease is frequently underdiagnosed due to atypical symptoms and its late presentation in females, and some studies even suggest a 10 to 12 years delay in appearance. Our prevalence of PAD in women was 3.8% compared to 7.9% in males, perhaps partly, again, due to the age group selection, needing further studies to observe prevalence in older female groups.

### 4.3. Risk Factors

The association of PAD with coronary artery disease is observed in 50% of patients and in 20% of patients with cerebrovascular disease [25,26], becoming therefore an important indicator of the coexistence of other cardiovascular disorders [27]. Up to 40% of subjects with PAD die from coronary artery disease and 10–20% by cerebral artery disease [10]. This risk is not only high in asymptomatic subjects but has a straight relationship among symptomatic individuals and severe PAD measured through ABI [28]. Overall, only less than 40% of subjects with PAD do not have concomitant coronary or cerebrovascular disease. Therefore, the association of PAD with other cardiovascular diseases is one of the main concerns of this disease. In our study, 13.6% of our subjects with PAD had an associated history of ischemic heart disease, and only a small few had a history of ischemic cerebrovascular disease.

The strongest predictor of PAD in those 65 years and younger has been associated with diabetes and smoking [3], although our study concurs with this from the male sex perspective. This was different in our study for females. After logistic regression analysis, we found an independent risk factor association between women who were smokers and had hypertension ORs of 3.3 (95% CI: 1.2–9.0) and 5.2 (95% CI: 1.6–17.3), showing a stronger association between smoking and hypertension than with diabetes in our female group.

Stratification of cardiovascular risk was also attempted through the REGICOR score to identify those who had a higher risk of developing a cardiovascular event in the following 10 years. Altogether, we found higher scores in males (*p* < 0.001) with a high-risk score (considered > 10%) in those with PAD at 14.8% versus those with non-PAD at 12.7%. Therefore, screening with ABI in these subjects we believe to be important for the early implementation of secondary cardiovascular prevention.

### 4.4. Limitations

It should be noted that this study is part of a larger pilot screening program evaluating abdominal aorta aneurysms [8], thus explaining the chosen method of selection. Nonetheless, in order to minimize any selection bias that could affect our results, we reviewed our study population with the 2018 Catalan Health Survey “Enquesta Catalunya de Salut 2018” (ESCA) [29], focusing on the population age group, 65–74 years of age, and to the data obtained from our primary care centers in 2018, focusing on 65 years of age. We looked into various risk factors among our participants and those of ESCA and our primary care centers shown previously. In general, our participants are similar, although some differences are to be expected due to the wider age group of the survey and different methods of obtaining participant data. All this could lead to some variations seen in our sample participants. Due to limitations in our resources, we were forced to focus on one specific age group in our study. Additionally, our participation rate was 43.1%, which is consistent with the participation rate of other screening programs performed in our region that have a wider broadcasting and media diffusion such as colon and breast cancer screening in Barcelona, which achieved participation rates of 43.6% and 54.7%, respectively [30]. Overall, we believe this is a correct representation of our population, although these limitations should be noted.

## 5. Conclusions

In summary, the overall prevalence of PAD in North Barcelona at 65 years old is still significantly low compared to other regions of Europe [10], and male subjects were still predominantly affected by this disease than females at this age.

Our results concur with the literature on this topic for our region [14,17] and outline the strong association between male smokers with diabetes and female smokers with hypertension, stressing the importance of our focus on these subgroups and perhaps emphasizing the need for ABI screening programs, as well as implementing secondary prevention if PAD is confirmed in these specific group of subjects. Additional follow-up studies would be interesting to further evaluate our initial results and differences in sex, as well as prevalence in older populations.

## Figures and Tables

**Figure 1 jcm-10-04467-f001:**
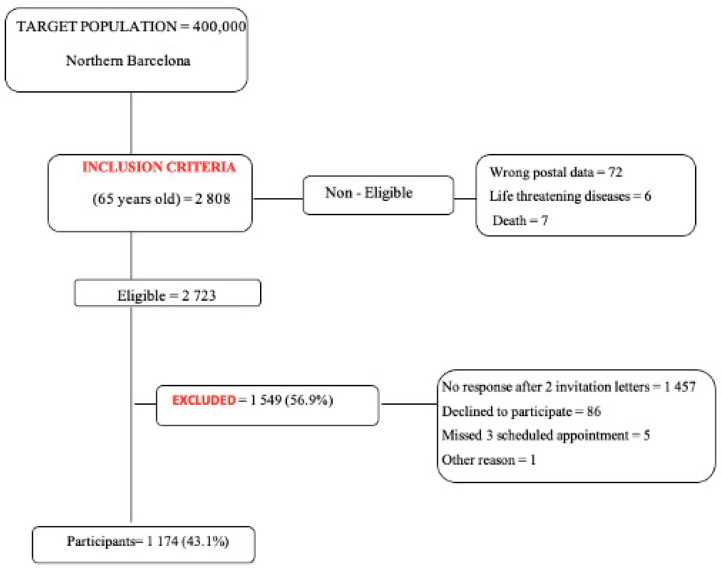
Flowchart of initial inclusion description.

**Figure 2 jcm-10-04467-f002:**
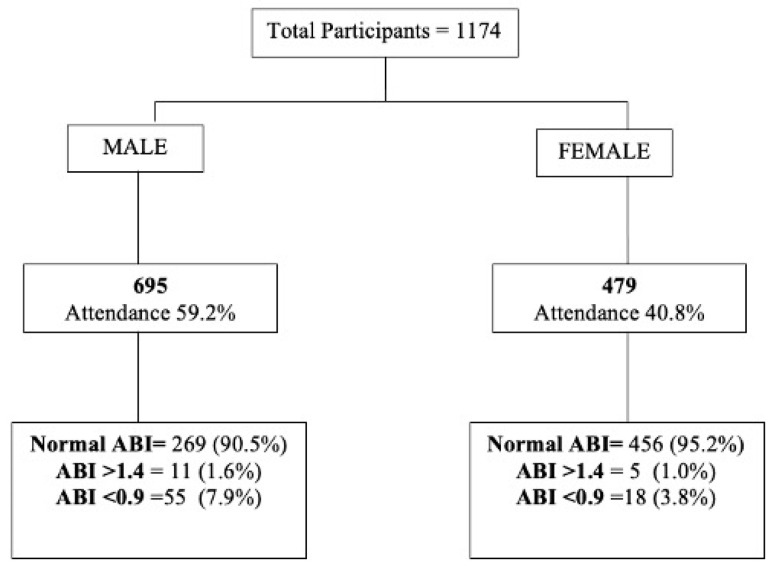
Flowchart screening results.

**Table 1 jcm-10-04467-t001:** Relationship between PAD and risk factors divided by gender. Logistic regression model with OR and CI 95% of the potential variables associated with PAD.

	Male					Female				
	Non-PAD (*n* = 640)	PAD (*n* = 55)	Total (*n* = 695)	OR * (95% CI: **)	*p* ^a^	Non-PAD (*n* = 461)	PAD (*n* = 18)	Total (*n* = 479)	OR (95% CI: **)	*p* ^a^
Non-smoker	237 (37.0%)	6 (10.9%)	243 (35.0%)	1	<0.001	320 (69.4%)	7 (38.9%)	327 (68.3%)	1	0.02
Former smoker	299 (46.7%)	29 (52.7%)	328 (47.2%)	3.8 (1.6–9.40)	0.003	96 (20.8%)	6 (33.3%)	102 (21.3%)	2.9 (0.9–8.7)	0.065
Active smoker	104 (16.3%)	20 (36.4%)	124 (17.8%)	7.6 (3.0–19.5)	<0.001	45 (9.8%)	5 (27.8%)	50 (10.4%)	5.1 (1.5–16.7)	0.007
**Cardiovascular risk factors n (%)**										
Diabetes	141 (22.0%)	20 (36.4%)	161 (23.2%)	2.0 (1.1–3.6)	0.017	43 (9.3%)	4 (22.2%)	47 (9.8%)	2.8 (0.9–8.8)	0.083
Dyslipidemia	293 (45.8%)	26 (47.3%)	319 (45.9%)	1.1 (0.6–1.8)	0.831	215 (46.6%)	10 (55.6%)	225 (47.0%)	1.4 (0.6–3.7)	0.459
Hypertension	340 (53.1%)	28 (50.9%)	368 (52.9%)	0.9 (0.5–1.6)	0.752	181 (39.3%)	12 (66.7%)	193 (40.3%)	3.1 (1.1–8.4)	0.026
Chronic Renal Disease	37 (5.8%)	3 (5.5%)	40 (5.8%)	0.9 (0.3–3.2)	0.920	8 (1.7%)	0 (0%)	8 (1.7%)	0	0.9
Cardiac Ischemia	59 (9.2%)	9 (16.4%)	68 (9.8%)	1.9 (0.9–4.1)	0.092	6 (1.3%)	1 (5.6%)	7 (1.5%)	4.5 (0.5–39.1)	0.177
Cerebrovascular Events	31 (4.8%)	3 (5.5%)	34 (4.9%)	1.1 (0.3–3.8)	0.840	16 (3.5%)	0 (0%)	16 (3.3%)	0	0.9
**Anthropometric measurements mean (SD)**										
Waist Circumference (cm)	102.4 (SD 11.0)	103.6 (SD 9.4)	102.5 (SD 10.9)	1.01 (0.99–1.04)	0.410	95.1 (SD 12.1)	102.4 (SD 15.9)	95.3 (SD 12.3)	1.05 (1.01–1.09)	0.018

Data are expressed as *n* (%) or mean (standard deviation); * OR: odds ratio; ** CI: confidence interval; **^a^** Wald test’s *p*-values of logistic regression.

**Table 2 jcm-10-04467-t002:** Multiple logistic regression analysis of PAD risk.

Male		Female	
**Variables**	OR (95% CI)	Variables	OR (95% CI)
**Non-smokers**	1	**Non-smokers**	1
**Former Smokers**	3.5 (1.4–8.7)	**Former Smokers**	3.1 (1.0–9.6)
**Active Smokers**	7.2 (2.8–18.6)	**Active Smokers**	5.2 (1.6–17.3)
**Diabetes**	1.8 (1.0–3.3)	**High blood pressure**	3.3 (1.2–9.0)

CI: confidence interval. OR: odds ratio.

**Table 3 jcm-10-04467-t003:** REGICOR results in general population separated by sex and PAD.

	Non-PAD			PAD		
	TOTAL (*n* = 972)	MALE (*n* = 543)	FEMALE (*n* = 429)	TOTAL (*n* = 41)	MALE (*n* = 27)	FEMALE (*n* = 14)
Low risk <5%	51.4% (500)	33.1% (180)	74.6% (320)	48.8% (20)	40.7% (11)	64.3% (9)
Moderate risk 5–9%	40.4% (393)	54.1% (294)	23.1% (99)	41.5% (17)	44.4% (12)	35.7% (5)
High risk >10%	8.1% (79)	12.7% (69)	2.3% (10)	9.8% (4)	14.8% (4)	0% (0)

Pearson’s chi-square analysis; 4 subjects without REGICOR.

**Table 4 jcm-10-04467-t004:** Means and proportions of selected characteristics in the screened population separated by sex.

	Total (*n* = 1174)	Male (*n* = 695)	Female (*n* = 479)
**Cardiovascular Risk Factors% (N)**			
Non-smokers ^a^	48.6% (570)	35.0% (243)	68.3% (327)
Active smoker	14.8% (174)	71.8% (124)	10.4% (50)
Former smoker	36.6% (430)	47.2% (328)	21.3% (102)
Relative’s history of AAA	3.0% (35) *	2.3% (16)	4% (19) *
First degree	74.3% (26)	87.5% (14)	63.2% (12)
Second degree	14.3% (5)	12.5% (2)	15.8% (3)
Diabetes mellitus ^a^	17.7% (208)	23.2% (161)	9.8% (47)
Dyslipidemia	46.3% (544)	45.9% (319)	47% (225)
Hypertension ^a^	47.8% (561)	52.9% (368)	40.3% (193)
Chronic renal disease ^a^	4.1% (48)	5.8% (40)	1.7% (8)
**Cardiovascular events % (N)**			
Cardiac ischemia ^a^	6.4% (75)	9.8% (68)	1.5% (7)
Cerebrovascular events	4.3% (50)	4.9% (34)	3.3% (16)
Intermittent claudication	3.8% (45)	4.5% (31)	2.9% (14)
**Anthropometric measurements % (N)**			
Waist circumference (cm) ^b^ (SD)	99.5 (12)	102.4 (10.9)	95.3 (12.3)
Mean BMI (kg/m^2^) (SD)	27.7 (4.3)	27.8 (3.8)	27.6 (4.9)
Normal weight ^a^ (BMI 18.5–24.9)	28.3 % (332)	24.2% (168)	34.2% (164)
Overweight (BMI 25–30)	46.3% (543)	51.4% (357)	38.8% (186)
Obesity (BMI >30)	25.5% (299)	24.5% (170)	26.9% (129)

* 4 of these, unknown degree; ^a^ *p* < 0.001 Pearson’s chi-square; ^b^
*p* < 0.001 Student’s *t*-test; Percentage (n). Mean and standard deviation.

**Table 5 jcm-10-04467-t005:** Characteristics of our study participants and participants of Catalunya Health Survey 2018 (ESCA) and primary care centers of our region of study separated by sex.

	Our Study Participants (65 Years Old)			ESCA 2018 (65–74 Years Old)			Primary Care Centers (65 Years Old)		
	Male (*n* = 695)	Female (*n* = 479)	Total (*n* = 1 174)	Male (*n* = 206)	Female (*n* = 224)	Total (*n* = 430)	Male (*n* = 1653)	Female (*n* = 1994)	Total (*n* = 3647)
Active smoker	17.8% (15.0–20.7)	10.4% (7.7–13.2)	14.8% (12.8–16.9)	18.0% (12.7–23.2)	8.0% (4.5–11.6)	12.8% (4.5–11.6)	20.8% (18.8–22.7)	13.4% (11.9–14.9)	16.8% (15.5–17.9)
Diabetes mellitus	23.2% (20.0–26.3)	9.8% (7.2–12.5)	17.7% (15.5–19.9)	27.2% (21.1–33.3)	20.1% * (14.8–25.3)	23.5% * (19.5–27.5)	23.1% (21.0–25.1)	12.8% * (11.4–14.3)	17.5% (16.2–18.7)
Dyslipidemia	45.0% * (42.2–49.6)	47.0% (42.5–51.4)	46.3% (43.5–49.2)	34.5% * (28.0–41.0)	46.0% (39.5–52.5)	40.5% * (35.8–45.1)	39.4% * (37.1–41.8)	41.7% * (39.5–43.8)	40.7% * (39.1–42.3)
Hypertension	53.0% (49.2–56.7)	40.3% (35.9–44.7)	47.8% (44.9–50.6)	50.5% (43.7–57.3)	48.2% * (41.7–54.8)	49.3% (44.6–54.0)	53.8% (51.4–56.2)	62.9% * (60.8–65.1)	58.8% * (57.2–60.4)
Chronic renal disease	5.8% (4.0–7.5)	1.7% (0.5–2.8)	4.1% (3.0–5.2)	5.8% (2.6–9.0)	6.3% * (3.1–9.4)	6.1% * (3.8–8.3)	4.1% (3.1–5.0)	2.3% (1.6–2.9)	3.1% (2.5–3.6)
Cardiac ischemia	9.8% (7.6–12.0)	1.5% (0.4–2.5)	6.4% (5.0–7.8)	7.3% (3.7–10.8)	2.2% (0.3–4.2)	4.7% (2.7–6.6)	8.1% (6.8–9.4)	1.8% (1.2–2.4)	4.7% * (3.9–5.4)
Cerebrovascular events	4.9% (3.3–6.5)	3.3% (1.7–5.0)	4.3% (3.1–5.4)	5.8% (2.6–9.0)	2.7% (0.6–4.8)	4.2% (2.3–6.1)	3.1% (2.3–3.9)	1.4% (0.9–1.9)	2.1% * (1.7–2.6)
Overweight (BMI 25–30)	51.4% (47.7–55.1)	38.8% (34.5–43.2)	46.3% (43.4–49.1)	46.6% (39.8–53.4)	42.0% (35.5–48.4)	44.2% (39.5–48.9)	38.2% * (35.8–40.5)	30.9% * (28.9–33.0)	34.2% * (32.7–35.8)
Obesity (BMI >30)	24.5% (21.3–27.7)	26.9% (23.0–31.0)	25.5% (23.0–28.0)	27.2% (21.2–33.3)	17.4% * (12.4–22.4)	22.1% (18.2–26.0)	27.7% (25.6–29.7)	29.5% (27.5–31.5)	28.7% * (27.2–30.2)

Values are presented as percentages (95% Confidence Interval); Pearson’s chi-square analysis; * Difference in values among both studies.

## Data Availability

Not applicable.

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
