# Peer review of "Prevalence of Peripheral Arterial Disease and Associated Vascular Risk Factors in 65-Years-Old People of Northern Barcelona"

_jcm, 2021, doi:10.3390/jcm10194467_

Round 1

Reviewer 1 Report

The study is well designed and the paper is well written.

The results are not new in tems of risk factors for PAD. The idea of a screening programm for vascular diesease in elderly people is very important.

The reviewed study is absolutely relevant for the diagnostic of vascular diseases as it analyzed the value of prophylaxis and screening tests.

The results are not unexpected, but the conclusions are relevant.

Reviewer 2 Report

This study explored the prevalence of PAD and risk factors associated with PAD in the 65-year-old elderly living in northern Barcelona. This study showed a prevalence of PAD of 6.2%, and the incidence was higher in men than in women. Independent risk factors according to gender were smoking and diabetes in men, and smoking and hypertension in women. Interestingly, the authors attempted to increase the validity of the PAD-related risk factors presented in this study compared to other large-scale cohorts.

1. Why did the authors select only 65-year-olds as the study subjects? Please describe in detail in the introduction.

2. Did the authors define active smokers as those who smoked even one cigarette within 1 year of study initiation? Is the severity of smoking not considered in active smokers?

3. What covariates were adjusted for in multivariate logistic regression? How were the covariates determined?

4. How was the interaction effect of risk factors in multivariate logistic regression analysis?

5. In Table 5, which groups showed significant differences?? Was post-hoc analysis performed?

6. In the footnote of the table, it should be clearly stated which statistical analysis was performed for each.

7. The authors should describe in detail and clearly using subheading in Method.

8. ABI is an important variable in this study. In patients with ABI <1.0, was retesting done to improve the accuracy of PAD diagnosis? How did the authors deal with the ABI data for patients with irregular heartbeats and heart rates >100?

9. How was inter-rater reliability between investigators for ABI?

10. This study showed that the risk factors for PAD differs by gender. The authors should perform a mediation analysis to verify that gender has a mediating effect.

11. The authors suggested that “the presence of DM in male smokers increases the risk of PAD” or “the presence of HTN in female smokers increases the risk of PAD.” Does the risk of PAD increase as the burden of risk factors increases?

Reviewer 3 Report

I propose a slight correction of the title: Instead of "Prevalence of peripheral arterial disease and associated vascu-2 lar risk factors in 65-years-old of northern Barcelona" the better statements could be :Prevalence of peripheral arterial disease and associated vascu-2 lar risk factors in 65-years-old people of northern Barcelona" or "Prevalence of peripheral arterial disease and associated vascu-2 lar risk factors in 65-years-old patients of northern Barcelona"

The Introduction is very superficial and it does not provide a sufficient background for the presented research it should present more details.

It is not clear why one of the inclusion criteria is residence of participants for at least 6 months in the referred area.

The results of statistical analysis are presented only for data in table IV (partly). In my opinion the rresults of statistical analysis should be presented for all analysed variables.

The references are relatively old, as only 7 of 25 references have been published in last 5 years - some more actual references should be included in revised version of the article.

Round 2

Reviewer 2 Report

My concerns have not been completely resolved. For the following questions, provide additional statistical data in the author's response. Comment 11 indicates whether an increased number of risk factors is associated with an increased risk of PAD.

4. How was the interaction effect of risk factors in multivariate logistic regression analysis?

9. How was inter-rater reliability between investigators for ABI?

10. This study showed that the risk factors for PAD differs by gender. The authors should perform a mediation analysis to verify that gender has a mediating effect.

11. The authors suggested that “the presence of DM in male smokers increases the risk of PAD” or “the presence of HTN in female smokers increases the risk of PAD.” Does the risk of PAD increase as the burden of risk factors increases?
